# Pituitary Adenylate Cyclase-Activating Polypeptide (PACAP) and Sudden Infant Death Syndrome: A Potential Model for Investigation

**DOI:** 10.3390/ijms242015063

**Published:** 2023-10-11

**Authors:** Dénes Tóth, Gábor Simon, Dóra Reglődi

**Affiliations:** 1Department of Forensic Medicine, University of Pécs Medical School, Szigeti út 12, H-7624 Pécs, Hungary; gabor.simon@aok.pte.hu; 2Department of Anatomy, HUN-REG-PTE PACAP Research Team, Centre for Neuroscience, University of Pécs Medical School, Szigeti út 12, H-7624 Pécs, Hungary; dora.reglodi@aok.pte.hu

**Keywords:** PACAP, neuropeptide, sudden infant death syndrome, animal model

## Abstract

Sudden infant death syndrome (SIDS) represents a significant cause of post-neonatal mortality, yet its underlying mechanisms remain unclear. The triple-risk model of SIDS proposes that intrinsic vulnerability, exogenous triggers, and a critical developmental period are required for SIDS to occur. Although case–control studies have identified potential risk factors, no in vivo model fully reflects the complexities observed in human studies. Pituitary adenylate cyclase-activating polypeptide (PACAP), a highly conserved neuropeptide with diverse physiological functions, including metabolic and thermal regulation, cardiovascular adaptation, breathing control, stress responses, sleep–wake regulation and immunohomeostasis, has been subject to early animal studies, which revealed that the absence of PACAP or its specific receptor (PAC1 receptor: PAC1R) correlates with increased neonatal mortality similar to the susceptible period for SIDS in humans. Recent human investigations have further implicated PACAP and PAC1R genes as plausible contributors to the pathomechanism of SIDS. This mini-review comprehensively synthesizes all PACAP-related research from the perspective of SIDS and proposes that PACAP deficiency might offer a promising avenue for studying SIDS.

## 1. Introduction

According to the San Diego definition, the term sudden infant death syndrome (SIDS) refers to ‘the sudden unexpected death of an infant <1 year of age, with onset of the fatal episode apparently occurring during sleep, that remains unexplained after a thorough investigation, including performance of a complete autopsy and review of the circumstances of death and the clinical history’ [1]. During the first year of life, SIDS is the leading cause of post-neonatal mortality, underlining an important public health priority [2,3]. In several countries, the implementation of risk reduction public health campaigns with simple and low-cost recommendations, such as back sleeping, in the late 1980s and early 1990s resulted in a decrease in SIDS rates. However, the progress in reducing SIDS rates has slowed. Recent SIDS rates vary by country, although they can be strongly influenced by factors such as geographical location, climate, and ethnicity [4,5]. The lack of consistent terminology and the inconsistent reporting practices make monitoring mortality trends difficult [6,7].

Although epidemiological and clinical studies have identified various intrinsic and extrinsic risk factors for SIDS, the underlying mechanisms remain unclear. SIDS causes are thought to be complicated and multifactorial. Based on the nowadays most widely accepted theory for the pathogenesis of SIDS, called the ‘triple-risk model’ (TRM), SIDS occurs when an infant with a latent intrinsic vulnerability (e.g., genetic background, impaired cardiorespiratory and/or autonomic responses, dysfunction of sleep–waking state organization) is exposed to an exogenous trigger event (e.g., unsafe sleeping environment, tobacco smoke, cold or hot environment) during a critical developmental period of homeostatic control [8]. According to TRM, all three elements must be present for death. Case–control studies remain the cornerstone of SIDS research because randomized clinical trials are not feasible due to ethical concerns. Case–control studies, on the other hand, can only reveal relationships between specific factors and SIDS; causation cannot be deduced. Numerous investigations have focused on various physiological aetiologies that may lead to infant vulnerability. However, none of the studies have provided a satisfactory explanation [9]. TRM has been shown to be highly valuable in structuring experimental methods for studying SIDS. Despite our improved knowledge of SIDS pathophysiology, an expansion of the studies on physiological/pathophysiological pathways and genetic factors that may increase vulnerability to SIDS is required. Furthermore, future research must concentrate on the interaction of these pathways with known environmental and other behavioural risk factors. Animal models can be beneficial for investigating the underlying mechanism of SIDS; however, there is no known in vivo model that adequately reflects the genetic, developmental, and environmental risk factors related to SIDS discovered in previous epidemiological and clinical studies. Animal models, albeit not providing a definitive explanation, do offer reliability to some hypotheses concerning the aetiology of SIDS and might help clarify (at least in part) the discrepancies between certain parts of the literature findings [10,11].

Pituitary adenylate cyclase-activating polypeptide (PACAP), a member of the secretin/glucagon/growth hormone-releasing hormone/vasoactive intestinal peptide superfamily, is an evolutionarily highly conserved multifunctional and pleiotropic neuropeptide with well-known anti-apoptotic, anti-inflammatory, and antioxidant effects and has two functionally active isoforms: PACAP38 [12] and PACAP27 [13]. The residue containing 38 amino acids (PACAP38) is the predominant form in mammals and represents 90% of all PACAP in the body. PACAP acts on G-protein-coupled receptors. PACAP and vasoactive intestinal peptide (VIP) share two nonspecific receptors, namely VPAC1 and VPAC2, while the PAC1 receptor (PAC1R) is selective to PACAP. PACAP and its receptors are broadly expressed in the nervous system and peripheral organs. Nonspecific receptors are more related to peripheral actions, while PAC1R is abundant in the central nervous system and is associated with neurotrophic and neuroprotective effects [14]. The molecular signalling of PACAP is highly complex and can exhibit variations depending on the specific tissue or cell type. PACAP receptors belong to the class of secretin-like G-protein-coupled receptors. The main signalling pathways activated by PACAP receptors are the following: (I) Adenylate cyclase (AC)/3′,5′-cyclic adenosine monophosphate (cAMP) pathway: PACAP receptors (both VPAC1-2 and PAC1R) exhibit a preference for coupling with Gαs proteins, which results in the activation of AC and subsequently the production of cAMP. The build-up of intracellular cAMP also triggers the activation of protein kinase A (PKA), which contributes to the neurotrophic functions of PACAP. These functions include stimulating neuronal stem cells, promoting neuroblast differentiation, regulating cerebellar development, modulating synaptic plasticity, promoting neuronal survival, and neuritogenesis. PKA may activate the extracellular signal-regulated kinase (ERK) signalling pathway, facilitating cell proliferation. Interestingly, PACAP can promote cell proliferation through a cAMP/ERK-dependent mechanism, which operates independently from PKA [14,15]. (II) Phospholipase C (PLC)/calcium pathway: PAC1R activates PLC through Gq signalling, causing an increase in intracellular calcium levels. This activation of the PLC/calcium pathway by PAC1R contributes to various biological functions, including cell migration, neuroplasticity, and the release of neurotransmitters and neurohormones [14,16]. (III) cAMP response element-binding protein (CREB) pathway: CREB can be activated by multiple kinases, including those that are components of the downstream signalling pathways of the PAC1R or VPAC1-2 receptors. The activation of CREB by PACAP enhances cell differentiation and facilitates the secretion of neurotransmitters and neurohormones in the hypothalamus [14,15,16]. (IV) G-protein-independent pathways: Little is known about the physiological role of the G-protein-independent pathways. A specific splice variant of PAC1R, PAC1nR-hop1, can activate phospholipase D and downstream signalling pathways. Additionally, internalized PAC1R receptors can trigger G-protein-independent signalling pathways, such as the phosphatidyl inositol 3 kinase/Akt and mitogen-activated protein kinase pathways [17,18,19,20]. The distribution of PAC1R as well as VPAC1-2 receptors within a specific tissue shows significant variability, and through alternative splicing, numerous variants can also be formed. Furthermore, the activation of the extensive signalling pathways outlined above explains the pleiotropic effects of PACAP [14]. The PAC1R-mediated pathways are summarized in Figure 1.

Based on the results of in vitro and in vivo experiments, the high translational value of PACAP became obvious. The description of the effects and localization of PACAP and its receptors in human tissues was followed by the detection of PACAP levels in several human body fluids. PACAP has been found to be taking part in a wide range of physiological processes, including autonomic and cardiorespiratory regulation [21,22,23], and dozens of recent studies have investigated the presence and changes of this peptide in various human pathological processes [24,25], including different types of traumatic injuries [26,27]. The hypothetical role of PACAP (and other endogenous neuropeptides) in the development of SIDS was first mentioned in 2004, as PACAP has a known role in homeostasis, including metabolism and thermo- and cardiorespiratory regulation [28]. It was also observed in animal studies that lack of PACAP or PAC1R is associated with a high neonatal mortality rate during the vulnerable period of development, equivalent to the age of risk for SIDS in humans [29,30]. Based on experiments with knockout animals, it was also evident that PACAP and/or PAC1R deficiency alone is not necessarily fatal. However, the disturbances in the PACAPergic system, such as ‘intrinsic vulnerability’, appear to interact with exogenous trigger events to increase the susceptibility to infant death [31]. A comprehensive and evolving systems-level model of SIDS identified PACAP as one of the candidate molecules in SIDS via the role of cardiopulmonary regulation [32]. A recent review focusing on the phenotypes of cardiorespiratory failure, apnoea, and early post-natal death and their relationship to SIDS identified—among others—PACAP and PAC1R genes as strong candidates for being involved in the pathomechanism of SIDS [33]. In the present mini-review, we summarize the findings of PACAP-related research from the point of view of SIDS and propose that PACAP deficiency may serve as a valuable model for studying SIDS.

## 2. In Vivo Data

### 2.1. Knock Out Mouse Models

#### 2.1.1. PACAP Knock Out Mice

In vivo experiments with PACAP-null mice showed a relatively high neonatal mortality rate within an age range that resembles the “critical period” when most SIDS deaths occur [29]. Gray and colleagues conducted a study investigating the influence of PACAP on early postnatal lipid and carbohydrate metabolism. Their findings indicated that while some PACAP-null mice pups died suddenly without weight loss, most of them perished after several days due to an inability to gain weight, leading to eventual weight loss. A histological analysis revealed cardiac intracellular lipid accumulation, indicating a potential cardiovascular event in the sudden death cases. Liver and skeletal muscle samples also exhibited lipid accumulation. Notably, PACAP-null mice showed substantially increased serum cholesterol and triglyceride levels and significantly lower blood glucose levels than their littermates during fasting [34]. The absence of PACAP resulted in inadequate heat production in PACAP-null mice subjected to prolonged, yet mild, cold stress. PACAP’s role in thermoregulation may involve norepinephrine and epinephrine release from the adrenal medulla or norepinephrine from nerve endings in brown adipose tissue, mediated through multiple PAC1 receptor signalling pathways. Brown adipose tissue is a primary heat source in neonates and rodents, triggered by norepinephrine release, which activates hormone-sensitive lipase (HSL) and uncoupling protein 1 (UCP1) in brown adipocytes. This process breaks down stored triglycerides into free fatty acids, generating heat via UCP1, a specialized protein that uncouples mitochondrial respiration, releasing energy as heat. Interestingly, there were no differences in the amount of brown adipose tissue between wild and PACAP-null animals. However, the PACAP-null mice exhibited upregulated HS lipase and UCP1 mRNA compared to the wild-type controls. The PACAP-null mice displayed significantly reduced levels of norepinephrine and its precursor, dopamine, in postganglionic nerve terminals within brown adipose tissue. This suggests that the absence of PACAP inhibits dopamine and norepinephrine production during prolonged cold stress [35]. Hamelink et al. observed that mice lacking the PACAP gene experienced more pronounced hypoglycaemia than wild-type mice, both after an intraperitoneal bolus injection of insulin and in response to an overnight fast. As a result, the PACAP knockout mice were unable to survive metabolic stress. Immunohistochemical staining for the enzymes responsible for catecholamine synthesis, phenylethanolamine *N*-methyltransferase (PNMT) and tyrosine hydroxylase (TH), showed no discernible distinctions compared to wild-type mice. Furthermore, under baseline conditions, the secretory function of the adrenomedullary appeared unaffected in the PACAP-deficient mice, with plasma epinephrine levels mirroring those of wild-type mice during anaesthesia. Nevertheless, the PACAP knockout mice were unable to survive when exposed to metabolic stress, experiencing notably more pronounced hypoglycaemia than their wild-type counterparts, both after an overnight fasting and following an intraperitoneal insulin injection. PACAP seems to play a pivotal role in immediately triggering the activation of tyrosine hydroxylase, facilitating the necessary increase in catecholamine biosynthesis for sustained epinephrine secretion during metabolic stress [36]. Shintani and colleagues noticed that PACAP-null mice, which succumbed before reaching 20 days of age—an age that might be comparable to the “critical period” during which most SIDS deaths occur—exhibited significant weight loss approximately two days prior to their demise [37]. Cummings et al. found that PACAP-null mice exhibited higher neonatal mortality, mainly due to respiratory control defects. The lack of PACAP resulted in reduced baseline minute ventilation and impaired ventilatory responses to hypercapnia and hypoxia. Additionally, under anaesthetic-induced hypothermia, PACAP knockout mice suffered from prolonged apnoea preceding atrioventricular block. The authors’ conclusion posits that PACAP may be a central respiratory regulator. This hypothesis stems from the activation of PAC1R, which can trigger the cAMP/PKA pathway. This pathway has been identified as having a role in controlling respiration by influencing the excitability of respiratory units within inspiratory neurons in the pre-Bötzinger complex and hypoglossal neurons. Furthermore, it seems to play a part in modulating the excitability of expiratory neurons in the medulla [38]. Arata et al. examined the respiratory activity in PACAP-null, heterozygous, and wild-type mice under control, hypoxic, and hypercapnic conditions. Their findings revealed that a significant proportion of the PACAP-deficient mice died at the weaning age (4 weeks). These mice displayed lower baseline respiratory activity and abnormal responses to hypoxia, while their response to hypercapnia mainly remained normal. Intriguingly, the P7 PACAP-null mice underwent respiratory arrest in response to hypoxia. Real-time PCR and histology further demonstrated that the hypofunction of the medullary catecholaminergic A1/C1 neurons in PACAP-null mice is responsible for causing the blunted responses to hypoxia that may be involved in the SIDS-like phenotype of PACAP-null mice [39]. A proteomic investigation unveiled altered expression levels of various proteins in intact mice lacking endogenous PACAP. Notably, the PACAP-knockout mice exhibited a significant downregulation of proteins associated with oxidative stress and antioxidant defence. Moreover, substantial differences in glycosylation enzymes were observed between the PACAP-null animals and wild-type controls, with malate dehydrogenase 1, enolase 2, aldolase 1, phosphoglycerate mutase 1, and pyruvate kinase showing downregulation, while ATP synthase was upregulated [40]. Barett and co-workers investigated the influence of PACAP on the autonomic response to heat stress in neonatal mice. They found that the PACAP knockout mice exhibited a significantly reduced increase in skin temperature, and their respiratory rate was nearly four times lower compared to wild-type controls. Tidal volume and minute ventilation were significantly decreased in the PACAP-null animals during heat stress, and their apnoea durations were slightly longer than baseline. Additionally, PACAP-null mice showed a significantly lower increase in heart rate in response to heat stress. They concluded that the PACAP knockout neonates had attenuated heat responses, likely due to decreased sympathetic responses to heat compared with the wild controls [41].

#### 2.1.2. PAC1R Knock Out Mice

Jamen et al. reported a 60% mortality rate among PAC1R-null mice pups within the four weeks after birth [30]. The significance of PAC1R-mediated signalling in maintaining normal pulmonary vascular tone during early postnatal life was demonstrated by Otto et al. In the advanced stages, corticosterone, triglyceride, free fatty acid, ketone body, and lactate levels were notably elevated in the serum of the PAC1R-null mice, accompanied by the onset of fatty liver and fatty degeneration in cardiac muscle cells. The right ventricles of these mutant animals exhibited significant dilation, with larger cardiomyocytes in the right ventricular region. The echocardiography findings concluded that the PAC1R-deficient mice experienced selective right heart failure. The elevated systolic right ventricular pressure in the PAC1R-deficient mice suggested increased resistance in the right ventricular outflow tract, and there was evidence of heightened muscularization in small pulmonary artery vessels, indicative of pulmonary hypertension. In summary, the PAC1R-null mice developed pulmonary hypertension, resulting in right heart failure and subsequent mortality during the second postnatal week. This research left an open question regarding whether the PAC1R-deficient mice suffered from primary pulmonary hypertension or if pulmonary hypertension was a consequence of primary alveolar hypoxia [42]. Barett et al. examined the role of the PAC1 and VPAC2 receptors in the cardiorespiratory response to acute hypoxia during neonatal development. In PAC1R-null mice, they observed blunted respiratory rate, tidal volume, and minute ventilation responses to hypoxia, persisting throughout the challenge. These mice exhibited elevated expired CO_2_ at baseline, but the response was blunted during hypoxia. Furthermore, their post-hypoxic cardiorespiratory recovery was impaired, characterized by an elevated respiratory rate, minute ventilation, expired CO_2_, and heart rate, along with decreased apnoea frequency and duration and reduced ventilatory efficiency [43]. One year later, in a follow-up study, Barett et al. investigated the role of the PAC1R and VPAC2 receptors in the cardiorespiratory response to hypercapnia during neonatal development. Compared to the wild-type controls, blunted respiratory rates and minute ventilation responses to hypercapnia were observed in the PAC1R-deficient group. These effects persisted throughout the duration of the challenge. Additionally, impaired post-hypercapnic recovery of heart rate was also detected in this group [44].

### 2.2. Other Observations

In their study, Huang et al. assessed the stress effects of acute and repeated intermittent hypercapnic hypoxia, mimicking rebreathing in the prone position, and nicotine exposure on PACAP and PAC1R protein expression in the medulla of developing piglets. The study revealed that one-day intermittent hypercapnic hypoxia led to reduced PACAP expression in the dorsal motor nucleus of the vagus (DMNV), the nucleus of the solitary tract (NTS), and the gracile nucleus (GN), along with decreased PAC1R expression in the NTS. Surprisingly, these changes were not sustained when the exposures were repeated. Nicotine exposure, on the other hand, did not affect PACAP expression but decreased PAC1R expression in the DMNV. In the DMNV, a potential interaction between the β2 nicotinic acetylcholine receptor (nAChR) subunit and PAC1R has been hypothesized. This hypothesis is grounded in observations that PACAP enhances the release of acetylcholine at presynaptic terminals, augments synaptic transmission excitability, elevates the affinity and magnitude of current signals mediated by nAChRs, and influences downstream signalling pathways, including the activation of PKC and the reduction in catecholamine production. Acute intermittent hypercapnic hypoxia with a pre-exposure to nicotine did not affect PAC1R expression, but PACAP expression decreased in the GN [45]. In a mouse model of pre- to postnatal cigarette smoke exposure, a research group investigated PACAP and PAC1R expressions in nuclei related to respiratory regulation. Although no changes were observed in PACAP expression due to smoke exposure, PAC1R expression increased in the NTS, cuneate nucleus (CUN), nucleus of the spinal trigeminal tract (NSTT), and inferior olivary nucleus (ION). The authors proposed a hypothesis that this upregulation of PAC1R expression in their P20 mice might be mediated through the MAPK pathway, as this pathway is known to regulate PAC1R promoter activity [46]. Shi et al. conducted experiments using transgenic models and viral techniques. The selective deletion of PACAP in neurons of the retrotrapezoid nucleus (RTN) in mice led to increased apnoea and diminished CO_2_-stimulated breathing, further exacerbated by ambient temperature changes. Restoring PACAP expression corrected these observed breathing deficits. Similarly, PAC1R deletion in the pre-Bötzinger complex produced similar changes and suppressed PACAP-evoked respiratory stimulation in that region. The researchers identified PACAP gene expression in the RTN/parafacial respiratory group, with low levels during the late embryonic period, followed by a sharp increase in the days after birth, stabilizing breathing during the transition from the intrauterine environment to air breathing. The PACAP expression gradually diminished from the early postnatal stages into adulthood. They concluded that well-timed PACAP expression provides an important supplementary respiratory drive immediately after birth and supports breathing during a particularly vulnerable period in life [47].

## 3. Human Data

Cummings et al. investigated the PACAP gene (*ADCYAP1*) in 92 SIDS cases and 92 unrelated race- and sex-matched controls. The analysis revealed the identification of 11 PACAP variants in both groups. While no statistically significant associations were found in the Caucasian groups, a non-synonymous single nucleotide polymorphism in exon 2 of the PACAP gene (rs2856966) showed a significant association with SIDS in the African American subset. In this locus, the G allele and GA genotype were increased more than three-fold in the SIDS cases compared to the controls [48]. Barett et al. examined the PAC1R gene (*ADCYAP1R1*) in 96 SIDS cases and 96 unrelated controls, detecting 61 known and 5 novel gene variants. While no strong association between variants in *ADCYAP1R1* and SIDS was found, the following potential race-specific associations were identified: the simultaneous occurrence of variant ‘q’ in *ADCYAP1R1* and the rs1893154 variant in *ADCYAP1* among African Americans and variant ‘i’ in *ADCYAP1R1* and the rs8192597 or rs2856966 variants in *ADCYAP1* among Caucasians [49]. Huang et al. investigated the neuroanatomical distribution and localization of PACAP and PAC1R in the human infant brainstem and hippocampus (HC) in relation to SIDS and potential risk factors. They observed significantly higher PACAP expression in the dorsal raphe nucleus of the SIDS cases compared to the non-SIDS infants and significantly lower PAC1R expression in the arcuate nucleus (AN) of the SIDS victims compared to the non-SIDS infants. Additionally, higher PACAP expression was detected in the cornu ammonis (CA) 2 and 3 regions of the HC in SIDS infants who bed-shared and in the subiculum of SIDS victims who died between March and August (cold months). On the other hand, lower PAC1R expression was found in the AN for SIDS infants who were not immunized and in the vestibular nucleus for those with a positive report of a recent upper respiratory tract infection. Furthermore, lower PAC1R immunosignal was detected in the CA2 region of the HC in the SIDS victims who died between March and August. A lower PAC1R expression trend was also observed in the cold-month SIDS case sub-group in the hypoglossal nucleus and in the DMNV [50].

## 4. Discussion

SIDS is multifactorial, reflecting complex interactions between genetic and exogenous factors; however, the pathogenesis remains unknown. Even though epidemiological and clinical studies have revealed several intrinsic and extrinsic risk factors for SIDS, the underlying mechanisms remain unknown. Since randomized clinical trials are not feasible due to ethical considerations, case–control studies remain the cornerstone of SIDS research. Case–control studies, on the other hand, can only identify relationships between certain factors and SIDS; no causation can be determined. Given this difficulty, animal models can be beneficial when studying the underlying mechanism of SIDS. However, no known in vivo model fully reflects the genetic, developmental, and environmental risk factors associated with SIDS identified in prior epidemiological and clinical investigations [8,9,10,11].

PACAP appears to be an ancestral molecule, as evidenced by its conservation over 700 million years of evolution in terms of both the length and sequence identity of nucleotides and amino acids in mammals and other tetrapods. The high level of conservation observed in PACAP strongly implies its vital functions for survival. Multiple lines of evidence support the notion that PACAP may play essential roles in the development of the nervous system and various organs, including the liver, pancreas, and adrenal glands in mammals [51,52]. Mice lacking PACAP or PAC1R have a higher rate of neonatal mortality compared their wild-type littermates. Based on the current data, the PACAPergic system appears to be implicated in multiple loci that may contribute, either directly or indirectly, to the development of sudden infant death (Figure 2), suggesting that alterations in its function, whether genetic, in expression, or translation, could potentially render the infant ’vulnerable’ in this context.

Dozens of in vivo data highlight the role of PACAP in metabolism [53], thermal regulation [54,55], cardiovascular adaptation [56,57], central and peripheral control of breathing [58,59,60], responses to stress [55,61,62,63,64], sleep–wake regulation [65,66] and immunohomeostasis [67]. The lessons we have learnt from PACAP or PAC1R knockout models exhibit multiple similarities with human data from SIDS case studies: In PACAP or PAC1R deficiency cases, high neonatal mortality rates occur within an age range resembling the ‘critical period’ when most SIDS deaths occur [29,30].PACAP deficiency was found to be associated with impaired lipid and carbohydrate metabolism and an impaired response to metabolic stress [34,36]. For years, it has been known that metabolic diseases can occasionally manifest as SIDS, and human studies indicate that metabolic disorders might be responsible for 1% to 2% of SIDS cases [68,69]. These conditions can lead to severe cardiac failure, shock, cardiac arrest, or acute metabolic crises. Among metabolic disorders, fatty acid oxidation disorders are the most common culprits behind SIDS, and they can often appear with minimal or no preceding clinical symptoms [70,71]. These fatty acid oxidation disorders may be associated with lipid accumulation in various tissues, including skeletal muscle and the liver. Regarding carbohydrate metabolism, genetic deficiencies in the hepatic glucose-6-phosphatase system can lead to fasting hypoglycaemia and, hence, the risk of SIDS. Low hepatic glucose-6-phosphatase catalytic subunit 1 (G6PC1) activities have been previously observed in some full-term SIDS infants. In preterm SIDS infants, there is an indication of disrupted development, as failures in the postnatal activation of G6PC1 expression have been reported [72,73]. The significant downregulation of proteins linked to oxidative stress and antioxidant defence in PACAP-knockout mice [40] underscores that PACAP deficiency influences homeostasis and metabolic activity across multiple domains.Thermoregulatory disturbances have been noted in PACAP-deficient mice, where the absence of endogenous PACAP has led to insufficient heat production, primarily due to inadequate norepinephrine stimulation of brown adipose tissue during prolonged but mild cold stress [35]. This altered thermogenesis of PACAP-null mice has also been confirmed by another study [54]. They display hypometabolism and hypothermia under restrained conditions, partially compensated by increased locomotor activity under unrestrained conditions. PACAP is known to inhibit GABAergic neurons in the hypothalamic preoptic area and median preoptic nucleus, which leads to increased body temperature. The absence of this inhibitory mechanism could be the reason behind the decreased temperature and deficient thermoregulation in mice lacking PACAP, although the involvement of other brain structures has also been suggested in this mechanism [54]. During infancy, a hypothesis suggests that brown adipose tissue could play a role in the onset of sudden infant death syndrome (SIDS). This hypothesis is based on the idea that thermal stresses and disrupted thermogenesis are factors in this condition. However, it is important to note that human studies have not confirmed this hypothesis [74].In the PACAP knockout mice, there was also evidence of impaired autonomic responses to heat stress, characterized by reduced sympathetic responses. Specifically, the PACAP-null animals exhibited a significantly decreased respiratory rate, tidal volume, and minute ventilation during heat stress, along with slightly longer apnoea durations [41]. Infants are particularly vulnerable to heat stress because their temperature regulation mechanisms are still developing. The interplay between thermal stress and the body’s protective homeostatic responses can lead to potentially life-threatening situations, especially during sleep. It is known that respiration is highly influenced by thermoregulation, so thermal stress can have significant effects on the characteristics of respiratory control. Exposure to heat can negatively affect the autonomic nervous system, potentially disrupting the drive for cardiorespiratory function and hindering the ability to awaken when a vital system is compromised [75]. A study involving sleeping preterm neonates observed that even small thermal loads are associated with reduced overall heart rate variability [76]. Autonomic control also appears to be impaired in SIDS victims [77,78]. Anatomical abnormalities within the intermediolateral nucleus of the spinal cord have been documented in 60% of SIDS cases [79]. This nucleus houses PACAPergic preganglionic sympathetic neurons, which are crucial for regulating cardiorespiratory responses to heat and various physiological stressors. Notably, it is susceptible to pathological changes when exposed to maternal cigarette smoke, a significant risk factor associated with SIDS.In mice, PAC1R deficiency has been associated with pulmonary hypertension, which can lead to right heart failure and subsequent death within the second postnatal week [42]. Human data regarding abnormalities in pulmonary vessels observed in cases of SIDS are limited and somewhat controversial. Early observations noted increased muscularity in the pulmonary circulation of SIDS victims [80]. Bradley and colleagues reported a case of SIDS where pulmonary artery thickening and associated pulmonary hypertension played a critical role in the death of a 23-day-old infant [81]. Another study found that the mean relative medial thickness of the alveolar wall arteries did not differ between SIDS cases and age-matched control cases. However, within the SIDS group, the thickest alveolar wall arteries were significantly more likely to be males and premature births [82].The PACAP-null mice exhibited higher neonatal mortality primarily due to defects in respiratory control. They displayed reduced baseline minute ventilation and impaired ventilatory responses to hypercapnia and hypoxia. When exposed to hypothermia, these mice experienced prolonged apnoea, and, notably, the P7 PACAP-null mice had respiratory arrest in response to hypoxia [38,39]. PAC1R deficiency also led to impaired cardiorespiratory responses to both hypoxia and hypercapnia. This included blunted respiratory rate and minute ventilation responses to both hypoxia and hypercapnia, along with impaired post-hypoxic and post-hypercapnic cardiorespiratory recovery [43,44]. An investigation into the impact of acute and repeated intermittent hypercapnic hypoxia, simulating rebreathing in the prone position, on PACAP and PAC1R protein expression revealed that intermittent hypercapnic hypoxia led to a decrease in both PACAP and PAC1R expression in the brainstem nuclei responsible for respiratory regulation [45]. Another study indicated that precisely timed PACAP expression in the pre-Bötzinger complex and the RTN/parafacial respiratory group plays a crucial role in providing additional respiratory drive immediately after birth and supporting breathing during a particularly vulnerable phase of life [47]. These findings align with human data since the central nervous system in specific brainstem regions within the ventrolateral medulla controls the cardiorespiratory system. It is worth noting that each of these brainstem regions has specialized functions in regulating breathing and heartbeat: the pre-Bötzinger complex governs inspiration, the post-inspiratory complex controls post-inspiratory activity, and a subset of the parafacial respiratory group manages active expiration [83]. TH immunoreactive catecholaminergic neurons within the ventrolateral medulla oblongata display delayed dendritic development in SIDS individuals [84]. Furthermore, TH immunopositivity experiences a significant decrease in the DMNV and the area reticularis superficialis ventrolateralis of the medulla oblongata in SIDS cases [85].Nicotine exposure resulted in decreased PAC1R expression in piglets in the DMNV, while in mice, an increase in PAC1R expression was observed in NTS, CUN, NSTT, and ION [45,46]. Pre- and/or postnatal exposure to cigarette smoke is a factor that increases infant vulnerability to SIDS, with more than 40 studies showing a positive association and risk ratios ranging from 0.7 to 4.85. This heightened risk of SIDS is likely attributed to the effects of nicotine exposure on autonomic control and arousal, affecting, among others, the same brainstem nuclei [86].

In relation to the reviewed literature data (Table 1) and the overlaps between human SIDS research data, we believe that PACAP or PAC1R deficiency acts as an intrinsic vulnerability that might interact with exogenous trigger events to increase susceptibility to infant death.

Interestingly, CD1 background PACAP knockout mice have emerged as a valuable animal model for the three-hit theory of depression. In this model, PACAP deficiency serves as the genetic background (hit 1, representing intrinsic vulnerability) and contributes to both early-life environmental challenges (hit 2) and late-life environmental stress (hit 3) [87].

Drawing upon the collective understanding of PACAP’s significance during development and its crucial role in adaptation and homeostatic regulation, coupled with insights gained from studying PACAP/PAC1R knockout mice, we consider these animals as promising candidates for an animal model of SIDS.

## Figures and Tables

**Figure 1 ijms-24-15063-f001:**
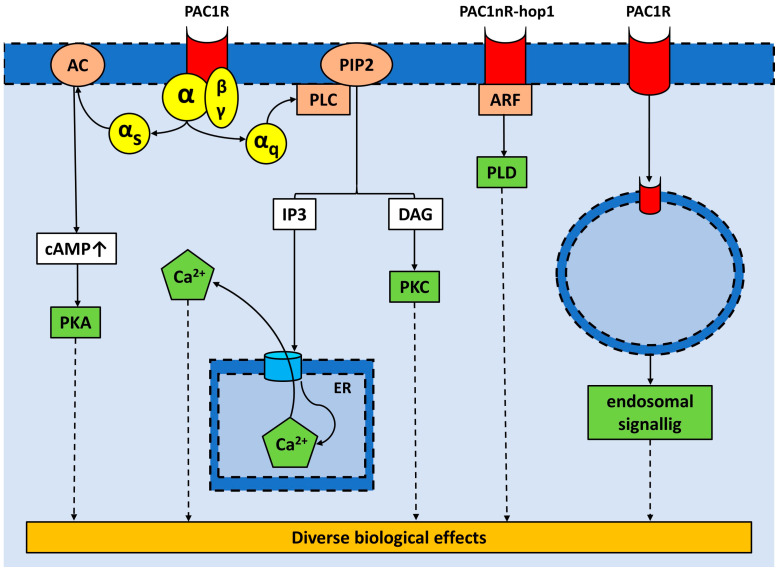
PAC1R-mediated signalling pathways. PAC1R exhibits a preference for coupling with Gαs, resulting in the activation of adenylate cyclase, followed by cyclic adenosine monophosphate production and subsequent activation of protein kinase A and other related downstream signalling pathways. The activation of PAC1R can also stimulate phospholipase C through Gq signalling, resulting in an intracellular calcium level increase and the activation of protein kinase C. Pathways independent from G proteins encompass the activation of phospholipase D and downstream signalling, as well as endosomal signalling pathways. (Abbreviations: AC = adenylate cyclase, ARF = adenosine diphosphate ribosylation factor, cAMP = cyclic adenosine monophosphate, IP3 = 1,4,5–inositol trisphosphate, PAC1R = PACAP type I receptor, PAC1nR-hop1 = PACAP type I receptor splice variant, PIP2 = phosphatidylinositol bisphosphate, PKA = protein kinase A, PKC = protein kinase C, PLC = phospholipase C, PLD = phospholipase D).

**Figure 2 ijms-24-15063-f002:**
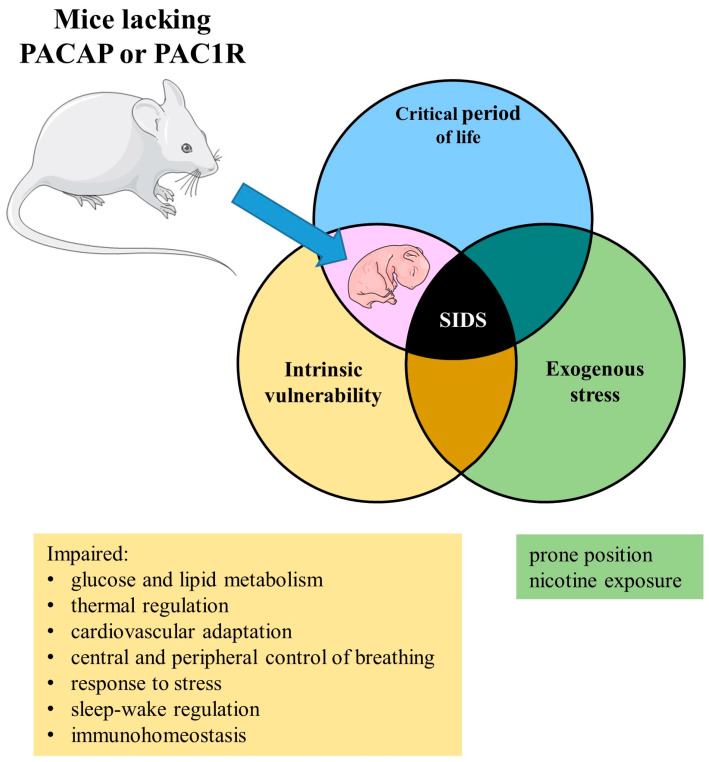
Anomalies in the PACAPergic system through the lens of the triple-risk model. Disturbances in the PACAPergic system make these mice susceptible (intrinsic vulnerability), particularly during the critical developmental stage, corresponding to the age range in which most human SIDS cases occur. Parts of the figure were drawn by using pictures from Servier Medical Art. Servier Medical Art by Servier is licensed under a Creative Commons Attribution 3.0 Unported License (https://creativecommons.org/licenses/by/3.0/, accessed on 1 September 2023). (Abbreviations: PACAP = pituitary adenylate cyclase-activating polypeptide; PAC1R = pituitary adenylate cyclase-activating polypeptide type 1 receptor; SIDS = sudden infant death syndrome.).

**Table 1 ijms-24-15063-t001:** The main findings of PACAP-related research from the point of view of SIDS. (Abbreviations: AN = arcuate nucleus; ATP = adenosine triphosphate; CO_2_ = carbon dioxide; CUN = cuneate nucleus; DMNV = dorsal motor nucleus of the vagus; DRN = dorsal raphe nucleus; GN = gracile nucleus; HC = hippocampus; HN = hypoglossal nucleus; ION = inferior olivary nucleus; PACAP = pituitary adenylate cyclase-activating polypeptide; PAC1R = pituitary adenylate cyclase-activating polypeptide type 1 receptor; RTN = retrotrapezoid nucleus; SIDS = sudden infant death syndrome; SNP = single nucleotide polymorphism; SUB = subiculum; NSST = nucleus of the spinal trigeminal tract; NTS = nucleus of the solitary tract; VEST = vestibular nucleus).

	**Condition(s)**	**Observations**	**Reference**	**Translational Value**
PACAP knock out mice	general	- high neonatal mortality rate	[29]	corresponds with epidemiological data related to human SIDS
	metabolism (lipid and carbohydrate)	- elevated serum levels of cholesterol and triglycerides; - lower blood glucose levels in respond to fasting	[34]	can be identified in specific subsets of human SIDS cases.
	thermal regulation (cold stress)	- inadequate heat production under prolonged but mild cold stress	[35]	not proved in human SIDS cases
	metabolism (carbohydrate)	- more profound hypoglycaemia both after an intraperitoneal bolus injection of insulin and in response to an overnight fast	[36]	can be identified in specific subsets of human SIDS cases.
	general	- marked weight loss during ∼2 days preceding their death	[37]	no human data
	respiratory and thermal (cold stress) regulation	- reduced the baseline minute ventilation and the ventilatory response to hypercapnia and hypoxia; - prolonged apnoea under anaesthetic-induced hypothermia	[38]	corresponds/similarities with data related to human SIDS
	respiratory regulation	- lower baseline respiratory activity and abnormal responses to hypoxia; - impaired catecholaminergic system in the medulla oblongata	[39]	corresponds/similarities with data related to human SIDS
	general	- downregulation in a group of proteins related to oxidative stress and antioxidant defence; - upregulation of ATP synthase	[40]	no human data
	respiratory and thermal (heat stress) regulation	- lesser increase in skin temperature and respiratory rate in response to heat stress; - decreased tidal volume and minute ventilation in heat stress; - longer apnoea in heat stress	[41]	corresponds/similarities with data related to human SIDS
PAC1R knock out mice	general	- high neonatal mortality rate	[30]	corresponds with epidemiological data related to human SIDS
	general	- pulmonary hypertension	[42]	controversial data in human studies
	cardiorespiratory regulation	- blunted respiratory rate, tidal volume, and minute ventilation responses to hypoxia; - impaired post-hypoxic cardiorespiratory recovery; - reduced ventilatory efficiency	[43]	corresponds/similarities with data related to human SIDS
	cardiorespiratory regulation	- blunted respiratory rate and minute ventilation responses to hypercapnia; - impaired post-hypercapnic recovery of heart rate	[44]	corresponds/similarities with data related to human SIDS
Other in vivo data	respiratory regulation (prone position, nicotine exposure)—piglet	- decreased PACAP expression in the DMNV, NTS and GN, and decreased PAC1R expression in the NTS after one day intermittent hypercapnic hypoxia; - decreased PAC1R expression in the DMNV after nicotine exposure; - decreased PACAP expression in the GN in case of acute intermittent hypercapnic hypoxia with pre-exposure to nicotine	[45]	corresponds/similarities with data related to human SIDS
	respiratory regulation (cigarette smoke exposure)—mice	- increased PAC1R expression in the NTS, CUN, NSTT, and ION	[46]	corresponds/similarities with data related to human SIDS
	respiratory regulation (transgenic mice and viral techniques)	- selective deletion of PACAP in the RTN resulted in increased apnoea and blunted CO_2_-stimulated breathing, and re-expression of PACAP corrected these deficits; - selective PAC1R deletion in the pre-Bötzinger complex, and re-expression showed similar changes	[47]	corresponds/similarities with data related to human SIDS
Human data	genetic study	- in the African-American subset, a non-synonymous SNP of the PACAP gene was identified, which associated significantly with SIDS	[48]	-
	genetic study	- potential associations between race-specific variants of PAC1R gene and SIDS were identified	[49]	-
	immunohistochemical study	- higher PACAP expression in the DRN; - lower PAC1R expression in the AN; - higher PACAP expression in the HC in case of bed shared SIDS victims; - higher PACAP expression in the SUB of SIDS victims who died in cold months; - lower PAC1R expression in the HC, HN and DMNV of SIDS victims who died in cold months; - lower PAC1R expression in the AN for those SIDS infants that were not immunized; - lower PAC1R expression in the VEST for those with a positive report of a recent upper respiratory tract infection	[50]	-

## Data Availability

Not applicable.

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
