# Peer review of "Pituitary Adenylate Cyclase-Activating Polypeptide (PACAP) and Sudden Infant Death Syndrome: A Potential Model for Investigation"

_ijms, 2023, doi:10.3390/ijms242015063_

Round 1
Reviewer 1 Report
In this review, the authors have reviewed and summarised data on PACAP and PAC1 in relation to SIDS incorporating both human and animal model data. It is well written, of good English, and easy to follow.
However, it is written as a literature review and although it seems to have covered core studies available, there is no clear critical analysis of the findings whereby the authors provide their input as to the strengths vs weaknesses of the studies they summarise. If the authors wish to undertake this task (which I understand is very time consuming and a whole different way of reviewing), this can be incorporated in Table 1 by providing a column titled “Strengths of findings”. Some examples of strengths to be indicated in this column would be “reproduced”, “supported by human SIDS data” etc. For example, PACAP -/- have lower blood glucose due to fasting- but how does this relate to SIDS? Have elevated serum levels of cholesterol and triglycerides been found in SIDS?
Other comments for correction
- Table 1 on pg 8/11 has the reference numbers missing in the last column (it currently only has “data” there).
- Lines 246-249 and Fig 1 - as the authors noted, a complete knockout of PACAP and PAC1 results in neonatal death. But SIDS is not a neonatal death. As such, wouldn’t it be best to suggest that altered function (be it genetic, expression, translation) would make the infant “vulnerable”; in effect, Fig 1 has 2 concerns for me:
1- The -/- on the rodent,
2- The arrow of the pup being in critical period of life
The change in PACAP/PAC1 is actually making the baby “vulnerable”. The authors have correctly verbalised this on line 261 but Fig 1 does not match this.
- Concluding paragraph- this needs more elaboration with the authors providing some tangible suggestions as to what further studies are required from these animal models to help “solve/explain” SIDS, or at the very least, which of the data they refer to , make them “promising candidates”. This links with my overall issue that there is no critical review of the data and as such, it is not clear how the data from these models have helped narrow the gap of understanding SIDS.
Author Response
In this review, the authors have reviewed and summarised data on PACAP and PAC1 in relation to SIDS incorporating both human and animal model data. It is well written, of good English, and easy to follow.
We would like to thank to the review for dedicating her/his time to thoroughly read the article and for providing valuable and constructive feedback. The insights and suggestions have been instrumental in improving the quality of our work. The changes made in the manuscript are visible alongside the Microsoft Word track changes function.
However, it is written as a literature review and although it seems to have covered core studies available, there is no clear critical analysis of the findings whereby the authors provide their input as to the strengths vs weaknesses of the studies they summarise. If the authors wish to undertake this task (which I understand is very time consuming and a whole different way of reviewing), this can be incorporated in Table 1 by providing a column titled “Strengths of findings”. Some examples of strengths to be indicated in this column would be “reproduced”, “supported by human SIDS data” etc. For example, PACAP -/- have lower blood glucose due to fasting- but how does this relate to SIDS? Have elevated serum levels of cholesterol and triglycerides been found in SIDS?
Completely in agreement with the reviewer's standpoint, we have significantly expanded the discussion chapter, where we summarized the results of the literature review of the PACAPerg system in bullet points and compared them with human SIDS research data. We also incorporated this information into Table 1 by creating a new column called "Translational value."
Table 1 on pg 8/11 has the reference numbers missing in the last column (it currently only has “data” there).
Thank you for pointing out this deficiency; we have added the missing reference numbers.
Lines 246-249 and Fig 1 - as the authors noted, a complete knockout of PACAP and PAC1 results in neonatal death. But SIDS is not a neonatal death. As such, wouldn’t it be best to suggest that altered function (be it genetic, expression, translation) would make the infant “vulnerable”;
We share the reviewer's viewpoint and have subsequently rephrased the ambiguous sentence.
Fig 1 has 2 concerns for me: 1-The -/- on the rodent, 2-The arrow of the pup being in critical period of life. The change in PACAP/PAC1 is actually making the baby “vulnerable”. The authors have correctly verbalised this on line 261 but Fig 1 does not match this.
We acknowledge the reviewer's concerns and have made revisions. Specifically, we have adjusted the figure to indicate that disruptions in the PACAPergic system render these animals susceptible, particularly during the critical life stage that aligns with the age range in which most human SIDS cases occur. As a result, we have repositioned the pup in the diagram accordingly. Please note that a new Figure has been added to the manuscript, so the referenced figure in the question has been changed to Figure 2. We have also supplemented the Figure 2 caption with an additional sentence to aid in the interpretation of the figure.
Concluding paragraph- this needs more elaboration with the authors providing some tangible suggestions as to what further studies are required from these animal models to help “solve/explain” SIDS, or at the very least, which of the data they refer to, make them “promising candidates”. This links with my overall issue that there is no critical review of the data and as such, it is not clear how the data from these models have helped narrow the gap of understanding SIDS.
We have conducted a comparative analysis of the literature pertaining to the PACAPergic system alongside the research findings on human SIDS in the discussion section. This comparative assessment is summarized in Table 1, with a particular emphasis on elucidating its translational significance. We hope that with these additions, we have successfully supported why we believe these animal models are promising candidates.
Reviewer 2 Report
The authors described that PACAP/PAC1R knockout mice are a potential model for investigation of SIDS. The authors well explained not only the background of PACAP related biology, but also the study results on literatures. However, the authors did not reveal any MOA on PACAP sgnaling, but just desribing the PACAP/PAC1 expression. To assess the validity of PACAP/PAC1R knockout mice, cell-based/signal-based evidence should be discussed in main part, even though there might be still not confident yet. The discussion should be assembled from in vivo and in vitro MOA studies. Indeed, if the authors can comment MOA things, this manuscript may be worth to publishing in the IJMS.
Author Response
The authors described that PACAP/PAC1R knockout mice are a potential model for investigation of SIDS. The authors well explained not only the background of PACAP related biology, but also the study results on literatures. However, the authors did not reveal any MOA on PACAP signalling, but just describing the PACAP/PAC1 expression. To assess the validity of PACAP/PAC1R knockout mice, cell-based/signal-based evidence should be discussed in main part, even though there might be still not confident yet. The discussion should be assembled from in vivo and in vitro MOA studies. Indeed, if the authors can comment MOA things, this manuscript may be worth to publishing in the IJMS.
We want to express our sincere gratitude for taking the time to review our manuscript and providing valuable comments and suggestions. Your input has been very helpful in improving the quality of our work. The changes made in the manuscript are visible alongside the Microsoft Word track changes function.
We extended the introduction section with a separate paragraph that contains the key molecular details of PACAP-mediated signalling. For completeness, we have also prepared a new figure (Figure 1) related to this.
In Section 2, where it was relevant, we provided detailed experimental results and, when available, also discussed the molecular background.
We have substantially enhanced the Discussion section by condensing the findings from the PACAPerg system's literature review into bullet points and conducting a comparative analysis with research data related to human SIDS. Additionally, we have integrated this data into Table 1, introducing a novel column labelled "Translational value."
We hope that we have correctly interpreted the reviewer's suggestions and have successfully addressed all the questions that arose.
Round 2
Reviewer 1 Report
All points have been addressed sufficiently.